# Identification and Characterization of Glucosyltransferase That Forms 1-Galloyl-*β*-d-Glucogallin in *Canarium album* L., a Functional Fruit Rich in Hydrolysable Tannins

**DOI:** 10.3390/molecules26154650

**Published:** 2021-07-31

**Authors:** Qinghua Ye, Shiyan Zhang, Nana Qiu, Linmin Liu, Wei Wang, Qian Xie, Qiang Chang, Qingxi Chen

**Affiliations:** 1College of Horticulture, Fujian Agriculture and Forestry University, Fuzhou 350002, China; tinayqh@163.com (Q.Y.); shiyan1909@163.com (S.Z.); qnn9605@163.com (N.Q.); llm08070406@163.com (L.L.); mykuangwen@hotmail.com (W.W.); xieq0416@163.com (Q.X.); 2Fujian Key Laboratory of Physiology and Biochemistry for Subtropical Plant, Fujian Institute of Subtropical Botany, Xiamen 361006, China; chyile_1@163.com

**Keywords:** polyphenols, hydrolysable tannins, UDP-glycosyltransferase, phylogenetic analysis, enzymatic catalysis

## Abstract

Hydrolysable tannins (HTs) are useful secondary metabolites that are responsible for pharmacological activities and astringent taste, flavor, and quality in fruits. They are also the main polyphenols in *Canarium album* L. (Chinese olive) fruit, an interesting and functional fruit that has been cultivated for over 2000 years. The HT content of *C. album* fruit was 2.3–13 times higher than that of berries with a higher content of HT. 1-galloyl-*β*-d-glucose (*β*G) is the first intermediate and the key metabolite in the HT biosynthesis pathway. It is catalyzed by UDP-glucosyltransferases (UGTs), which are responsible for the glycosylation of gallic acid (GA) to form *β*G. Here, we first reported 140 UGTs in *C. album*. Phylogenetic analysis clustered them into 14 phylogenetic groups (A, B, D–M, P, and Q), which are different from the 14 typical major groups (A~N) of *Arabidopsis thaliana*. Expression pattern and correlation analysis showed that *UGT84A77* (*Isoform0117852*) was highly expressed and had a positive correlation with GA and *β*G content. Prokaryotic expression showed that *UGT84A77* could catalyze GA to form *β*G. These results provide a theoretical basis on *UGTs* in *C. album*, which will be helpful for further functional research and availability on HTs and polyphenols.

## 1. Introduction

The role of secondary metabolites has been of interest to scientists for a long time. Hydrolysable tannins (HTs) are one category of useful secondary metabolites, which are a heterogeneous groups of water-soluble polyphenolic compounds of high molecular weight (500–3000 Daltons) with up to 20 hydroxyl groups [1]. Strong antioxidant and radical scavenging capacities make HTs play a role in the treatment of various diseases. The antioxidant activity and scavengers of hydroxyl, superoxide, and peroxyl radicals largely depend on their structure [2,3,4]; for example, an increase in anti-radical effects was observed with an increase in the degree of polymerization [5]. Modern medical research has shown that HTs have pharmacological activities against COVID-19 [6], bacteria [4], inflammation [7], nephropathy [8], diabetes [9], HIV, and many other pathologies [5,10,11]. Moreover, HTs are responsible for the astringent taste, flavor, stability, biological activity, etc. of many fruits [12,13]. HTs consist of multiple esters of gallic acid (GA) with glucose and products of their oxidative reactions, which can be generally classified into gallotannins and ellagitannins depending on the residues to which the hydroxyl group of glucose forms an ester linkage [5,14].

The synthesis of HTs depends on the catalysis of gallic acid UDP-glucosyltransferase (UGT). As the precursor of HT biosynthesis, 1-galloyl-*β*-d-glucose (syn. *β-*glucogallin*, β*G) is a vital intermediate and metabolite in the tannin biosynthesis pathway, where UGT is a key enzyme [14,15,16]. In *Arabidopsis thaliana*, UGTs responsible for glycosylation of flavonoids, benzoates, and terpenoids mainly exist in groups A, B, D, E, F, H, and L [17]. All known UGTs that form glucose esters with benzoates belong to group L [18], which is mainly composed of three subfamilies of UGT74s, UGT75s, and UGT84s, the UGT84s is the only subfamily with active *β*G formation function so far [19]. At present, *UGT* genes that control *β*G formation has been cloned and identified in *Vitis vinifera* [20], *Quercus robur* [21], *Eucalyptus camaldulens**is* [22], *Camellia sinensis* [23], *Fragaria* × *ananassa* [24], and *Punica granatum* [25,26], where they participate in the GA glycosylation reaction during the synthesis of HTs and procyanidins. These UGT enzymes were confirmed to be distributed in group L. However, the relationship between the degree of amino acid sequence identity and substrate specificity of the UGTs is highly complex. Furthermore, *UGTs* belong to polygenic families and may have different functional characteristics among different plants. Therefore, it is particularly important to identify and classify the UGT family and to screen and characterize UGTs with the potential function of forming *β*G. This research will help to adequately understand the mechanism of UGTs in the process of hydrolysis tannin synthesis.

*Canarium album* (Lour.) Raeusch. (*C. album*), also known as Chinese olive or Chinese white olive, is a plant in the Burseraceae family, unlike European olive (*Olea europaea* L.), which belongs to the Oleaceae family. *C. album*, a functional fruit rich in polyphenols, originates from southeast China and has been introduced to other tropical and subtropical Asian regions, including Vietnam, Malaysia, and Japan [27,28,29]. It contains many phenolic compounds [30,31,32,33,34], which are responsible for some pharmacological functions, such as antibacterial, antiviral, and anti-inflammatory activities [35,36,37]. The total phenolic content (TPC) reaches 1291 mg gallic acid equivalent (GAE)/100 g·FW [33], which is much higher than that of many other fruits, such as Chinese date, cranberry, sweetsop, apple, guava, strawberry, pomegranate, and persimmon [38,39]. The TPC (280.46 mg GAE/g·DW) of dried *C. album* fruits, usually used as a medicine, was higher than that of most common traditional Chinese herbs [34,40]. Furthermore, the crucial active polyphenolic component in *C. album* fruits was found to be HTs [41]. Quantitative analysis showed that HTs account for nearly 85% of TPC in *C. album* fruits, as the content of ellagitannins (823.8 mg GAE/100 g·FW) could reach 55.13% of the TPC [41], which was 2.3–13 times higher than that of berries with a higher content of hydrolyzed tannin (65–360 mg GAE/100 g·FW) [42,43]. According to these results, *C. album* is expected to be a typical material for the research and utilization of hydrolyzed tannin.

In order to deeper understand the biosynthesis and metabolism of HTs, we firstly identified and analyzed UGT gene family members in *C. album*, which is doubtless the first step towards their research and utilization. According to the expression pattern and correlation analysis, we identified UGT genes that may be involved in the generation of *β*G and characterized the UGT enzymes using prokaryotic expression. This study provides new insights into the important *UGT* genes involved in HT synthesis of *C. album*, which has great significance in research on its medicinal activity and therapeutic potential.

## 2. Results

### 2.1. Analysis of Total HTs, GA, and βG Contents in C. album

*C. album* fruit is a natural candidate for dietary supplements, and studies have shown that its pharmacological action is closely related to phenolic compounds, especially HTs [37,38,42,44,45]. During the growth and development of *C. album* fruit (Figure 1a), the dynamic change of total HTs contents was a down trend (Figure 1b). The HT content in unripened *C. album* fruit was higher. HTs remained at a high level (1279.26–5667.35 mg/100 g·FW) throughout fruit development and ripening.

Candidate peaks in the UPLC chromatogram were identified with the characteristic protonated/deprotonated molecular ions as the diagnostic ions. The retention times of GA (3.01 min) and *β*G (4.73 min) in the samples were the same as those in the standards. Through MRM mode, GA and *β*G were quantified with transitions of m/z 169.01→124.82 and 333.08→171.15, respectively (Appendix A). An attempt was made to monitor the dynamics of GA and *β*G content in *C. album* fruits, and their trends were generally similar, which exhibited an overall trend of first increasing and then decreasing (Figure 1c). Concentrations of GA and *β*G were higher in the early stages (10–50 DAF) than throughout fruit ripening, with the highest values at 20 DAF, reaching 19.55 μg/g·FW and 1096.58 μg/g·FW, respectively, which showed that the glycosylation of GA in *C. album* fruit should mainly occur at this stage. The synthesis of abundant *β*G, the precursor of HTs, was sufficient for tannin biosynthesis.

### 2.2. Identification and Phylogenetic Analysis of C. album UGT Family Members

By using Pfam searches and Local BlastX strategies, we obtained 176 and 176 potential UGT protein sequences from the *C. album* full-length transcriptome peptide database (PRJNA749395), respectively. The hits obtained from the two searches were combined, and the redundant sequences and sequences lacking the PSPG (plant secondary product glycosyltransferase) box were removed. The remaining potential UGT protein sequences were submitted to the HMMER website for verification. Finally, 140 *C. album* UGT protein sequences (CaUGTs) were identified after removing nonplausible proteins.

We constructed a phylogenetic tree with 278 UGTs from different species to clarify the phylogenetic group to which each CaUGT belonged (Appendix A). A total of 140 CaUGTs and 122 AtUGTs clustered into 14 typical major groups (A~N), and an outgroup (OG) [44,46] was used for phylogenetic analysis and classification. In addition, groups O and P were discovered in higher plants [45], while group Q was found only in *Zea mays* [47]; hence, four UGTs of *Prunus persica* [48], three UGTs of *Zea mays* [48], and one UGT of *Punica granatum* representing the O, P, and Q groups were also added. In addition, eight UGTs from other species involved in the glucosylation of GA to yield *β*G were also used to identify and classify candidate *C. album* UGTs.

The phylogenetic analysis results showed that *C. album* UGTs were clustered into 14 groups, A to Q, except for groups C, N, and O (Figure 1). CaUGT members were unevenly distributed in different groups. It was obvious that Group D had the largest number of UGTs in *C. album*, accounting for 25.71% of the total CaUGTs. The second group, L, accounted for 19.29%. Groups A, D, E, G, and L were the major groups in other higher plants [45]. This indicated that groups D and L might play more important roles in *C. album*. Interestingly, there were 13 members in group Q, which was a novel group found in maize with only seven UGT members [47].

To investigate the structural diversity of 140 members of the CaUGT family, we detected the distribution of conserved motifs of genes based on evolutionary relationships (Figure 2). Fifteen conserved motifs from 140 CaUGTs were identified by MEME online software; almost all CaUGTs contained motifs 1, 2, 3, 7, and 8, and all the CaUGTs contained motif 1, which included the complete PSPG box (positions 4–47) (Figure 3 and Appendix A).

### 2.3. Expression Patterns of Candidate UGTs in the Developmental Stages of C. album Fruits and Selection of Key CaUGT Genes

Based on RNA-seq data, we chose 37 *CaUGTs* downregulated during the developmental of *C. album* fruits as candidate *UGTs*, to further analyze the expression patterns. As shown in Figure 4, most of the candidate *CaUGTs* showed a high correlation with *β*G content, and their trends of the relative expression were similar to that of *β*G contents. However, only one *CaUGT* (*Isoform0117852*) was highly expressed in *C. album* fruit. Moreover, Isoform0117852 was came from group L, resulting in the potential to form *β*G. The correlation coefficient between *Isoform0117852* expression level and *β*G content was 0.923 (*p* < 0.001). Thus, the gene was speculated to be involved in the biosynthesis of *β*G. The correlation coefficient between *Isoform0117852* and GA was also high (0.900, *p* < 0.001), indicating that the gene may directly use GA as a substrate to catalyze *β*G formation. Therefore, *Isoform0117852* was selected as the key gene for HT synthesis in *C. album* for subsequent functional verification experiments. According to the UGT Nomenclature Committee (https://prime.vetmed.wsu.edu/resources/udp-glucuronsyltransferase-homepage/ugt-submission-instructions, accessed on 6 July 2021) [49], the *Isoform0117852* gene was named *UGT84A77*.

### 2.4. Real-Time Quantitative RT-PCR (RT-qPCR) Verification

We randomly selected 12 *CaUGTs* that downregulate expression agreed with the changes in GA and *β*G content, to experimentally validate further the expression patterns by RT-qPCR. RNA sampled from *C. album* fruit and gene-specific primers were designed (Appendix A) and synthesized for RT-qPCR validation. As shown in Figure 5, RT-qPCR data indicated the same expression tendency as the RNA-seq data. The results showed that the expression pattern of RNA-seq was reliable. Therefore, we selected *UGT84A77* (*Isoform0117852*) as a potential gene that can catalyze the synthesis of *β*G for further study.

### 2.5. The Full-Length CDS Analysis of UGT84A77

The full-length CDS of *UGT84A77* was 1509 bp, encoding 503 amino acids, as shown in Appendix A. ExPasy tools (http://web.expasy.org/protparam/, accessed on 1 April 2020) predicted a molecular weight of 56.079 kDa with an isoelectric point of 5.28. The sequence information of *UGT84A77* was submitted to the NCBI database with GenBank accession number is MZ048740. Pfam online software annotation indicated that the protein contained a conserved UDPGT domain, in which the highly conserved PSPG box consisted of 44 amino acids. The ‘HCGWNS’ residues might interact with the uracil moiety of UDP-glucose; the 22nd position was tryptophan (W), which could correctly locate UDP-glucose, and the last amino acid, glutamine (Q), at position 44 of the PSPG box, also contributed to determining the specificity for UDP-glucose as a donor, rather than galactose [45,50,51].

### 2.6. Subcellular Localization

To further explore the potential function of the *UGT84A77* genes, the subcellular localization of UGT84A77 was analyzed. The coding sequence of UGT84A77 without the stop codon was fused with the GFP reported gene. The *Agrobacterium* cultures with the recombinant vector and the 35S:GFP control were used to inject the *Nicotiana benthamiana* leaf epidermal cells. As revealed by confocal microscopy (Figure 6), the green fluorescence of the UGT84A77-GFP fusion protein was found to distribute in the nucleus, cytoplasm, and cytomembrane, similar to the signal of GFP protein of control.

### 2.7. Expression and Purification of Recombinant UGT84A77 Protein

SDS-PAGE was used to assess recombinant proteins (Figure 7). Compared with the empty vector pET28a, the recombinant vector pET28a-*UGT84A77* clearly expressed a protein band of nearly 57 kDa, which was consistent with the theoretically predicted molecular mass of UGT84A77 fused with the His-tag.

Ni-NTA agarose purification resin was selected to purify the target protein in the supernatant, and the protein was eluted with different concentrations of imidazole eluents (Figure 8). Low concentrations of imidazole could remove protein impurities, and high concentrations of imidazole were suited for target protein elution. The protein eluted at 180 mM and 200 mM imidazole elution had almost no miscellaneous band. As a result, the protein purified by 200 mM imidazole elution was quantified at a concentration of 425 μg/mL and used for subsequent enzyme tests.

### 2.8. Enzymatic Properties of Recombinant UGT84A77 Protein

To detect the catalytic activity of the UGT84A77 recombinant protein, GA and UDPG were used as substrates, and purified protein was added. The reaction products are shown in Figure 2A. Compared with the standards, GA and UDPG could be detected in both the UGT84A77 enzymatic reaction and CK. In contrast to CK, the UGT84A77 enzymatic reaction showed a newly produced substance at a retention time of 4.99 min. The ion fragment was analyzed in MRM mode and identified as *β*G, indicating that the purified UGT84A77 recombinant protein catalyzed the reaction of GA and UDPG to form *β*G. pH and temperature are important factors influencing enzymatic reactions [52]. The suitable conditions for the reaction of the UGT84A77 recombinant protein were slightly acidic, and the highest activity of the enzyme was observed at pH 5.0 (Appendix A). Moreover, the highest enzyme activity occurred at low temperature, specifically 10 °C (Appendix A).

The recombinant protein activities of UGT84A77 were measured at different concentrations of the two substrates at pH 5.0 and 10 °C. The Michaelis–Menten curves were plotted (Figure 9b,c), and the kinetic parameters were calculated (Table 1). In general, Km is an important parameter of enzyme activity, and its standard error is required to be less than 25% for the reliable calculation of enzyme kinetic parameters [53]. In this study, the errors of the two substrates were within this range, and the *p*-value of the repeated test showed that the fitting model was reliable, indicating that the calculated kinetic parameters were accurate. The Km of UGT84A77 was 108.90 ± 21.06 μM when GA was used as a substrate, and the Km of UGT84A77 was 193.30 ± 34.33 μM when UDPG was used as the substrate, indicating that UGT84A77 had a slightly higher affinity for GA than UDPG. The catalytic efficiency Kcat/Km for GA (34076.64 s^−1^·M^−1^) was 8.86 times higher than that for UDPG (3844.87 s^−1^·M^−1^), indicating that UGT84A77 favored GA as a substrate.

## 3. Discussion

### 3.1. The C. album Fruit, Especially the Young Fruit, Is Rich in HTs

*C. album* fruit is a natural candidate for dietary supplements. Studies have shown that its pharmacological action is closely related to phenolic compounds. Some pharmacological applications have been verified, such as anti-HIV [54] and antidiabetes treatment [37], the regulation of lipid metabolism [36], and the inhibition of colon carcinoma [55]. The crucial active component in *C. album* fruits was found to be HTs [41]. Many fruits with plentiful HTs have corresponding antioxidant activities, as the HTs are bioavailable and promote health [56]. The HT content in *C. album* is higher than that in most fruits and traditional Chinese herbs [35,39,40]. Although HTs were found in high concentrations in ripe *C. album* fruits (Figure 1b) [41], it seems that the HT content in unripened fruit was even higher. Notably, the contents of GA and *β*G had the highest values at 20 DAF, which showed that the glycosylation of GA in *C. album* fruit should mainly occur at this stage. Therefore, we suppose that the early developmental stages are the critical period of glycosylation in *C. album* fruits.

### 3.2. Analysis and Screening of the UGT Gene Family in C. album

Glycosylation is a physiological process in the biosynthesis of plant secondary metabolites that can increase molecular activity and diversity, and adjust cellular homeostasis [57,58]. Glycosylation can not only change the biological activity of plant secondary metabolites, but also increase the diversity of structure and function of natural products [59]. The UGT family is crucial for the glycosylation modification of fruit secondary metabolites [57]. Plant UGTs belong to the first family of 110 glycosyltransferase families (GTs, EC 2.4.x.y) from the CAZy database (http://www.cazy.org/GlycosylTransferases.html, accessed on 1 September 2019) [60,61]. There is a highly conserved PSPG-box (plant secondary product glycosyltransferase box) at the C-terminal of UGTs protein, which is a sugar donor binding site [45,50]. The UGT family has been extensively studied in different plants, but, until now, no further information about the UGT family in *C. album* has been available. This study involved a comprehensive investigation of the CaUGT family. We identified 140 *CaUGTs* in the full-length transcript of *C. album* (Figure 3). The total number of UGT family members in *C. album* was slightly higher than that in *A. thaliana* (122), *Pyrus bretschneideri* (139), *C. sinensis* (132), and *Prunus mume* (130), while it was less than that in *Malus domestica* (241), *Prunus persica* (168), *V. vinifera* (181), *Populus trichocarpa* (178), and *Z. mays* (147) [23,45,47,48,62,63]. Although the number of UGT family members varies among species, most plants have a relatively large UGT family, which may be related to the genome size of each species [45]. The expansion of *UGT* genes might be directive, resulting in diversity of plant secondary metabolite biosynthesis [64].

To date, UGT family members can be clustered into 18 distinct groups (A~R) with an outgroup (OG) [44]. There are some differences in conserved domains and conserved sites of different groups of UGT, the number of group members is different in different plants, which reflects the situation of gene replication and expansion, and related to gene the relationship of plant evolution [45]. *Arabidopsis* was divided into only 14 distinct groups (A-N) and OG [44,46]. The OG is mainly composed of AtUGT80 and AtUGT81 subfamilies, which are responsible for the formation of sterols and lipids. Unlike other plant UGTs, the PSPG motifs of UGT80 and UGT81 are less conserved, and the sequence homology with the non-plant UGT family is higher than that with other plant sequences, suggesting that they evolved before the radiation of plants from the other phyla [44,65]. The two new phylogenetic groups (O and P) were found later in apple, poplar, and other higher plants [45]. In addition, group Q has been observed in *Z. mays* [47]. Group R was not recognized in most previous reports, and its members were usually placed in group E, possibly due to limited taxon samplings [44,47]. In the UGT family in *C. album*, all 140 CaUGTs contained the conserved PSPG-box motif and were distributed in 14 groups (A, B, D~M, P and Q) (Figure 1). This indicated that the identification of CaUGT family members was reliable. The PSPG box was found in all UGTs from higher plant species and was closely associated with plant secondary metabolism functions [57,66]. The number of motifs in different phylogenetic groups was disparate. Furthermore, the distribution of some motifs displayed subgroup specificity. Differences in the motif distribution may be related to the functions of each group. Different groups have substrate specificity, such as the group L UGT preference for benzoates belong to group L [18]. However, nothing is known so far about the function of the some phylogenetic groups (C, I, J, K, M, N) [45]. The number of UGTs in *C. album* was larger than that in *Arabidopsis* mainly due to expansion within groups D, L, and Q. These results implied that UGTs in these groups may be critical for some type of metabolism in *C. album* fruit, such as safener-inducible protective activities [64,67], the formation galloylglucose esters by galloylation reactions [21,68], and *O*-glycoside-forming activities on flavonoids [44], although more detailed research is required. Surprisingly, there is a lack of members in groups C, N, and O, which suggests that some gene loss events have occurred in *C. album* [45,48]. Group C was also found to be missing in *G. hirsutum* [69]. Similarly, group N was found to be absent in the UGT family in pear [48]. Research showed that group N was obtained almost exclusively from monocots, and only one was observed in dicot plants [62].

The expression profiles of *CaUGTs* in the developmental stages of *C. album* fruit proved a different temporal-specific expression pattern. The high expression level in early developmental stages was likely to prepare for the formation of *β*G. Among them, *Isoform0117852* (*UGT84A77*) most likely promoted the formation of *β*G. It might also regulate the availability and biological activity of metabolic intermediates and play an important role in fruit development [57,66]. Hence, exploring the function of *UGT84A77* is necessary.

### 3.3. UGT84A77 Catalyzes the Formation of βG

Prokaryotic expression of the previously identified key genes showed that the *UGT84A77* gene could induce soluble target proteins. An enzymatic reaction in vitro indicated that UGT84A77 could catalyze the formation of *β*G from GA, but the activity of the enzyme varied with the buffer pH and reaction temperature. The main factors affecting enzyme activity are temperature, pH, ionic strength, and the concentration of substrate and enzyme [52].

Enzyme kinetic parameters can reflect the enzymatic properties of proteins. The enzymatic kinetic parameters of the UGT84A subfamily of different species were compared (Appendix A). Km is related only to the properties of the enzyme. The lower Km is, the more affinity it has with the substrate. Compared to other species, such as *Q. robur* UGT84A13 (420 μM), *C. sinensis* CsUGT84A22 (758.4 μM), and the *C. album*, UGT84A77 had a higher affinity for GA (Km = 108.90 ± 21.06 μM), which differed from that of *P. granatum* UGT84A24 (980 μM) by a factor of 8 but was close to that of *E. camaldulensis* UGT84A25a (168 ± 14 μM), implying that UGT84A77 has a similar function to UGT84A25a in *E. camaldulensis*. Kcat is the catalytic constant of the enzyme, indicating the ability of the enzyme to catalyze a specific substrate. The higher the value is, the faster the conversion rate of the substrate is. The Kcat value of UGT84A77 for the substrate was larger than that of most enzymes except for that of UGT84A25a and UGT84A26a, whereas the catalytic efficiency Kcat/Km of UGT84A77 for GA (34076.64 s^−1^·M^−1^) and UDPG (3844.87 s^−1^·M^−1^) was higher than that of other enzymes except UGT84A57 [20,21,22,23,25,70]. Kcat/Km, also known as the specificity constant, reflects both the affinity of the enzyme to the substrate and its catalytic capacity. Different enzyme kinetic parameters may occur in different species and different determination conditions, such as reaction conditions, enzyme purity, and substrate specificity [71].

Glycosylation reactions of UGT family genes have specific catalytic characteristics. In the UGT84A subfamily, hydroxyl cinnamate substrates are mainly preferred [72]. The similarity in substrate specificity and diversity may be due mainly to structural differences [71]. In this study, the recombinant protein UGT84A77 from *C. album* had a high affinity for GA, but the study of more extensive substrates and further enzyme kinetics reactions, as well as glycosylation-specific site recognition [73], are needed to more fully understand the substrate properties of UGT84A77.

## 4. Materials and Methods

### 4.1. Plant Materials and Sampling

The fruits of the *C. album* cultivar ‘Changying’, cultivated at the *C. album* plantation located in Minhou County, Fujian Province, China (26°13′ N, 119°02′ E, 127 meters altitude), were used as materials. Three healthy and approximately uniform trees, 15 years old and subjected to consistent management conditions, were selected for the experiment. *C. album* fruit samples were collected at 10, 20, 30, 40, 50, 60, 70, 90, 110, 130, 150, and 170 days after flowering (DAF) during the 2018 growing season, as shown in Figure 1a. At each developmental stage, representative fruits without visible blemishes or diseases were sampled from each tree. The amount of young fruit collected was determined according to the experiment, and 20–30 fruits were collected from each tree in the mature stages. The fruit pulp of all samples was isolated, immediately frozen in liquid nitrogen, and stored at −80 °C for further analyses.

### 4.2. Chemicals

GA, tannin acid and uridine 5′-diphosphate glucose (UDPG) standards were purchased from Yuanye Biotech Co., Ltd. (Shanghai, China); *β*G standard was obtained from Sigma-Aldrich (St. Louis, MO, USA); Folin–Ciocalteu’s reagent was obtained from Solarbio Science & Technology Co., Ltd. (Beijing, China); HPLC-grade methanol and acetonitrile were purchased from Merck (Darmstadt, Germany); and all other chemicals and reagents were of analytical grade and obtained from the Sinopharm Chemical Reagent Co., Ltd. (Shanghai, China) or TransGen Biotech (Beijing, China) unless otherwise stated.

### 4.3. Measurement and Quantification of Total HTs, GA, and βG

The determination of total HTs in *C. album* was performed by Folin–Ciocalteu’s method, with tannic acid as the standard [74]. The extracts were prepared according to the methods previously reported [41] with minor modifications. Samples of frozen flesh were ground to powder with liquid nitrogen, and 200 mg aliquots of ground samples were suspended in 800 μL of prechilled 80% methanol for metabolite extraction. The mixture underwent ultrasonication (KQ-300GDV Ultrasonic Instruments, Kunshan, China) for 30 min at 4 °C. After centrifugation for 10 min at 12,000 rpm and 4 °C, the supernatant was obtained. The procedure was then repeated. The supernatant was collected and vacuum concentrated using a centrivap (LABCONCO CentriVap, Kansas, MO, USA), and then 80% methanol was added to the concentrated supernatant to a final volume of 200 μL. The resulting solution was filtered at 0.22 μm and then determined and quantified by UPLC-MS/MS.

The UPLC-MS/MS system consisted of an ACQUITY ultrahigh-performance liquid chromatography system and an XEVO-TQS triple-quadrupole tandem mass spectrometer (Waters Corp., Milford, MA, USA) as the most selective analytical tool [75]. This system was used to detect the targeted metabolites, GA and *β*G. Chromatographic separation was conducted using a Merck ZIC-pHILIC column (100 mm × 2.1 mm, 5 μm) at 40 °C. Solvent A 5 mM ammonium formate in high-purity water and solvent B (0.1% formic acid in acetonitrile) were used as mobile phases. The gradient elution program started with 5% A, a linear gradient up to 41% A in 8 min, a return to the initial 5% A in 2 min, and 5% A maintained for 3 min. A constant flow rate of 0.4 mL/min was maintained the process. The loading volume was 2 μL. *β*G and GA were quantified by using the multiple reaction monitoring (MRM) mode with m/z transitions of 333.08 → 171.15 and 169.01 → 124.82, respectively. A full scan was performed with the electrospray ionization (ESI) source operated in positive mode for *β*G and negative mode for GA. The cone voltage and collision energy were 30 V and 15 V. The other main working parameters were as follows: capillary voltage 0.88 kV for *β*G and 1.27 kV for GA, desolvation (nitrogen) temperature 400 °C, ion source temperature 150 °C, cone gas (nitrogen) flow 150 L/h and desolvation gas (nitrogen) flow 800 L/h, and collision gas (argon) flow 0.13 mL/min. Data were acquired and statistically calculated by Masslynx version 4.1.

### 4.4. Identification and Characterization of C. album UGT Genes

In the *C. album* full-length Iso-Seq transcriptome database (PRJNA749395), we selected all putative *UGT* genes by Pfam (ID: PF00201) and retrieved the candidate UGT sequences relying on the conserved 44-amino-acid PSPG box [45,50]. We used the PSPG boxes of the following UGTs: UGT84A23 and UGT84A24 for *P. granatum* [25], UGT84A13 for *Quercus robur* [21], GT2 and GT5 for *F. × ananassa* [24], GT1, GT2 and GT3 for *V. vinifera* [20], UGT84A1 and UGT84A2 for *A. thaliana*, and UGT84A22 for *C. sinensis* [23] as queries to identify candidate sequences using the local BLAST function of BioEdit software against the whole *C. album* full-length Iso-Seq transcriptome peptide database with a cut-off *E-value* of 1 × 10^−5^. Subsequently, the hits obtained from the two searches were combined, and sequences that were redundant or lacked the PSPG domain were removed. Furthermore, sequence alignment was performed by MEGA 6.0 (MegaSoftware, Tempe, AZ, USA) to assess the candidate sequences for removing repeat sequences. Ultimately, the Hidden Markov Model (HMMER, http://www.ebi.ac.uk/Tools/hmmer/, accessed on 1 September 2019) webserver was used to confirm that each predicted *C. album* UGT protein sequence containing the UDPGT (PF00201.18) domain had significant hits, and candidate sequences identified as nonplausible proteins were discarded.

### 4.5. Phylogenetic and Classification Analysis of CaUGTs

A total of 140 candidate genes in *C. album* corresponding to the UGT family were obtained. To identify and classify these candidate UGTs, 122 UGT sequences of Arabidopsis and some known UGTs from other species were used for phylogenetic analysis by MEGA 6.0. Appendix A shows the information of UGTs for phylogenetic analysis. All these sequences were aligned using the ClustalW algorithm, and a phylogenetic tree was constructed by using the neighbor-joining (NJ) method. The bootstrap values were calculated with 1000 replicates [47,76].

The conserved motifs of the UGT proteins were identified using the online MEME procedure (http://meme-suite.org/, accessed on 1 April 2020) [77,78] with a maximum of 15 motifs per sequence. The phylogenetic tree of 140 CaUGTs and motifs was re-edited using TBtools software (http://www.tbtools.com/, accessed on 1 April 2020).

### 4.6. Expression Pattern and Correlation Analysis of Candidate CaUGTs

According to the FPKM value of RNA-seq data from our laboratory (PRJNA749395), we selected downregulated *CaUGTs* as the candidate genes. Then, we analyzed the expression patterns of candidate genes at four development stages (20, 40, 70, 110 DAF) of *C. album* fruit, and the correlation coefficients between their expression levels and the content of GA and *β*G. Spearman correlation analysis (*p* < 0.001) was performed on using SPSS19.0 software. According to the correlation, the key *CaUGTs* regulating *β*G formation in *C. album* fruit were screened further.

### 4.7. RNA Extraction and RT-qPCR

According to the instructions provided with the RNAprep Pure Plant Kit (Polysaccharides & Polyphenolics-rich) (Tiangen Biotech Co., Ltd., Beijing, China), total RNA was first extracted from frozen *C. album* flesh at four developmental stages (20 DAF, 40 DAF, 70 DAF, 110 DAF). The RNA quality was analyzed by agarose gel electrophoresis and quantified using a Nanodrop 2000 spectrophotometer (Thermo Scientific, Wilmington, DE, USA). Approximately 1.0 μg of total RNA was used for first-strand cDNA synthesis using FastKing gDNA Dispelling RT SuperMix (Tiangen) following the supplier’s manual.

Transcriptional profiles of 12 CaUGT genes at *C. album* fruit development stages were verified by RT-qPCR. Specific primer pairs are given in Appendix A. RT-qPCR was performed on a LightCycler 96 instrument (Roche, Basel, Switzerland) using SYBR Green to detect gene expression abundance according to the protocol of RealUniversal Color PreMix (SYBR Green) (Tiangen). We selected a stably expressed *β*-actin gene (ACTB) from the transcriptome, and, after verification, *CaACT7* was used as an internal reference in this research. The relative expression levels of the genes were calculated for developmental time points relative to the first sampling time point using the 2^−^^△△^^CT^ method [79]. Samples for RT-qPCR were run in three biological replicates with three technical replicates.

### 4.8. Cloning and Subcellular Localization

Using cDNA of *C. album* fruit as the template, the full-length CDS of *UGT84A77* was amplified by PCR using gene-specific primers (Appendix A). The CDS without stop codon of *UGT84A77* was cloned into the TOPO vector according to gateway technology and ligated to the expression vector pK7FWG2 by LR reaction, generating the UGT84A77-GFP fusion fragment. The recombinant vector pK7FWG2-*UGT84A77* was transformed into competent *Agrobacterium* GV3101 cells by the freeze-thaw method [80] for transient expression in the leaves of *Nicotiana benthamiana*. After incubation of 36–48 h, these leaves were observed the fluorescence signals by using the confocal microscope (TCS-SP8 Leica, Wetzlar, Germany). The excitation wavelength of GFP fluorescence observation was 488 nm, and the detection wavelength was 500–550 nm. The excitation wavelength of chloroplast auto-fluorescence observation was 488 nm, and the detection wavelength was 650–750 nm.

### 4.9. Prokaryotic Expression of UGT84A77

The CDS of *UGT84A77* was amplified by PCR, the amplified product was then purified, digested with *BamH I* and *Sal I*, and inserted into pET28a with the *UGT84A77* CDS fused with a His-Tag, yielding the construct pET28a-*UGT84A77*. PCR primer sequences are listed in Appendix A. The construct was transferred into *Escherichia coli* BL21 (DE3) (TransGen) cells for recombinant protein expression. A single colony was taken for overnight precultured and expanded culture at a ratio of 1:50. Four milliliters of the overnight culture solution was added to 200 mL Luria–Bertani (LB) liquid medium with 50 μg/mL kanamycin and incubated at 37 °C and 200 rpm until the OD_600_ value was 0.5–0.6, and 0.05 mM isopropyl-*β*-d-thiogalactopyranoside (IPTG) was added to induce the recombinant protein. No IPTG was used as a control (CK) and cultured at 16 °C and 200 rpm for 20 h. After induction, all the bacterial cells were centrifuged at 4 °C and 5000 rpm and resuspended in 20 mL PBS, and 1 mM phenylmethylsulfonyl fluoride (PMSF) was added. The cells were crushed by a freeze-thaw method, and the cleared supernatants (12,000 rpm for 30 min at 4 °C) were collected for column purification of the target protein. The purification procedure was performed as described in the instructions of the Ni-NTA-Sefinose^TM^ Column (Sangon Biotech Co., Ltd. Shanghai, China). Protein expression and purification were detected by SDS-PAGE (Solarbio Co., Ltd., Beijing, China). Protein concentration was determined by the Quick Start^TM^ Bradford Protein Assay (Bio-Rad Laboratories, Hercules, CA, USA).

### 4.10. Enzymatic Activity Assay of Recombinant Protein

The enzymatic reaction system consisted of 1 mM GA and 1 mM UDPG, 10 μg of purified recombinant protein, and 50 mM PBS (pH 7.0) to a final volume of 100 μL. The ingredients were mixed. After incubating for 1 h at 30 °C, 200 μL of methanol was added to stop the reaction, and the protein inactivated by boiling for 10 min was used as a negative control (CK). We also investigated the effects of pH ranging from 4.0 to 7.0 and temperature ranging from 0 °C to 40 °C on the enzymatic reaction to obtain the optimal conditions for the enzymatic reaction.

Enzyme reaction products were detected by UPLC-MS/MS. The enzyme reaction was extracted by the addition of 300 μL of ethyl acetate and thorough vortexing followed by centrifugation, and the supernatant was concentrated in a 2 mL centrifuge tube at low vacuum temperature until lyophilized and then redissolved with 150 μL 80% methanol. The supernatant was filtered through a 0.22 μm membrane and tested with a UPLC-MS/MS system under chromatographic and mass spectrometry conditions as described in the previous paragraph. The mobile phase gradient elution was as follows: initial, 5% A; 0–8 min, 5–41% A; 8–10 min, 41–60% A; 10–12.5 min, 60% A; 12.5–12.6 min, 5–60% A; 12.6–15 min, 5% A.

Enzyme kinetics analysis was carried out at different substrate concentrations (0, 10, 50, 100, 250, 500, 1000, 2000 μM). The specific activity of UGT84A77 (nkat/mg) was calculated as the nmol number of products per milligram of purified protein per second [68]. The kinetic constant (Km), maximum reaction rate (Vmax), enzyme catalytic constant (Kcat), and catalytic efficiency (Kcat/Km) were calculated according to the Michaelis–Menten model [22,81].

### 4.11. Statistical Analysis

Data were analyzed with Excel 2016 (Microsoft Corporation, Redmond, WA, USA), and SPSS 19.0 software (SPSS Inc., Chicago, IL, USA) was used for the statistical analysis. Data plots were generated by GraphPad Prism 7 (GraphPad Software, San Diego, CA, USA). One-way analysis of variance (ANOVA) and Tukey–Kramer multiple-comparisons test were used to determine significant differences. A level of significance of *p* < 0.05 (different lowercase letters) and *p* < 0.01 (different uppercase letters) were adopted. Correlation analysis was performed by Spearman correlation analysis (*p* < 0.0001). All experiments were independently repeated at least three times.

## 5. Conclusions

In summary, this study showed that *C. album* fruits were rich in HTs. The early developmental stages of *C. album* fruit are an important period of glycosylation in HT synthesis. We identified 140 CaUGTs and clustered them into 14 groups based on phylogenetic analysis. Expression pattern analysis and correlation analysis revealed that *UGT84A77* (*Isoform0117852*) was most correlated with *β*G formation. In an in vitro assay, *UGT84A77* was responsible for forming *β*G and had a high affinity for GA. This study provides a valuable reference for the metabolism and utilization of *β*G and HTs. It is of great significance to study the medicinal activity and therapeutic potential of functional fruits.

## Figures and Tables

**Figure 1 molecules-26-04650-f001:**
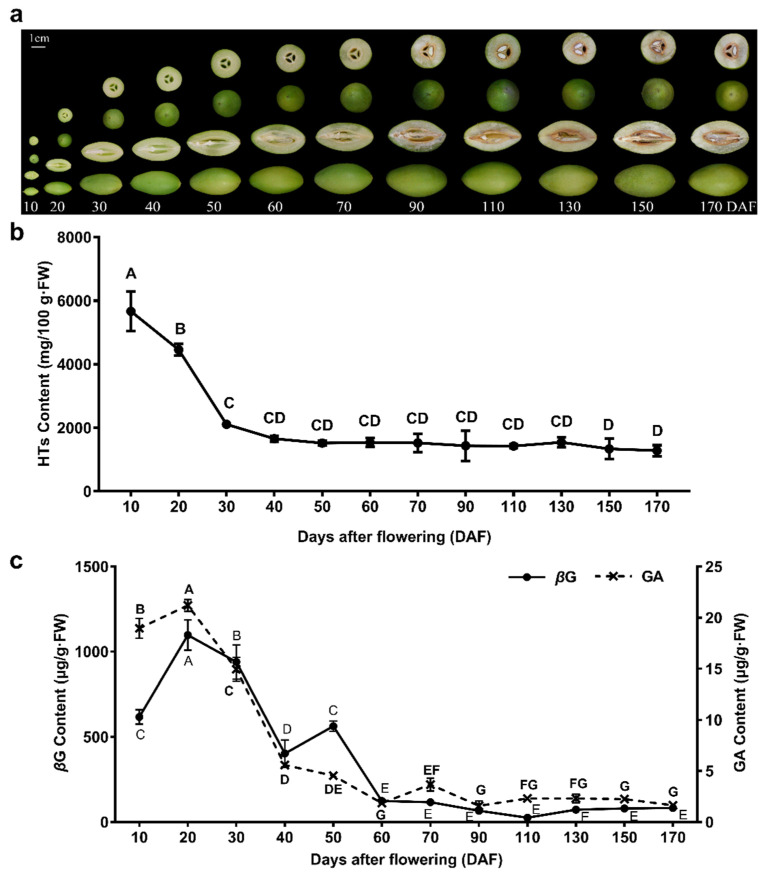
*C. album* fruit development and phenolic substance content. (**a**) The exterior and interior characteristics of *C. album* fruits in different development stages; (**b**) hydrolysable tannin (HT) content of *C. album* fruits in different development stages; (**c**) gallic acid (GA) and 1-Galloyl-*β*-d-glucose (*β*G) contents of *C. album* fruits in different development stages. Each data point is the average mean of five reactions ± SD. Different uppercase letters indicate significant differences at *p* < 0.01 levels, based on the Tukey–Kramer test.

**Figure 2 molecules-26-04650-f002:**
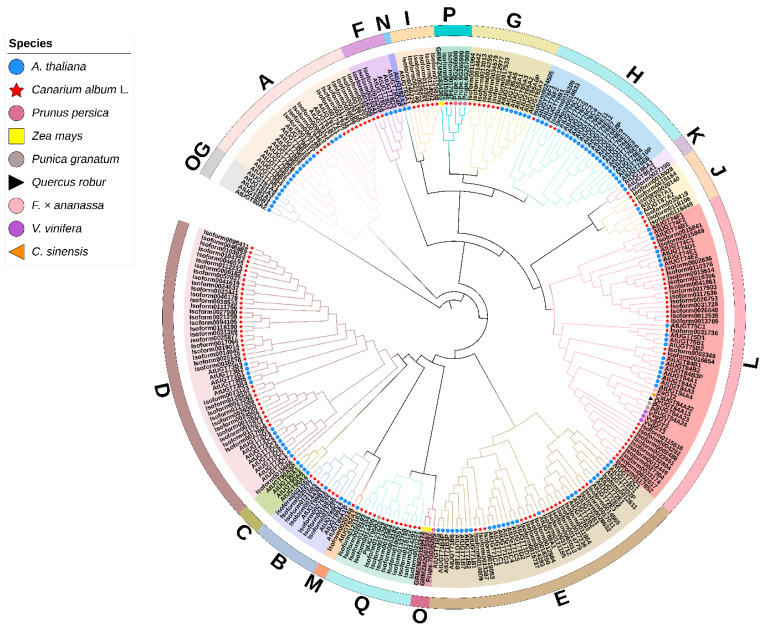
Phylogenetic analyses of *C. album* UDP-glucosyltransferase (UGT) family members. The phylogenetic tree was constructed in the MEGA 6.0 program using the Neighbour-Joining method, and the bootstrap value was set to 1000. A total of 278 amino acid sequences of different plant UGTs were used, including 140 *C. album* UGTs, 122 Arabidopsis UGTs, four *Prunus* UGTs, three *Z. mays* UGTs, one *Punica* UGT and eight other UGTs that have been functionally characterized, including UGTs from *Punica granatum*, *Quercus robur*, *F. × ananassa*, *V. vinifera*, and *C. sinensis*. The outermost letters indicate different groups, including groups A–Q and the OG. Their information was shown in Appendix A. Each group is highlighted in a different color.

**Figure 3 molecules-26-04650-f003:**
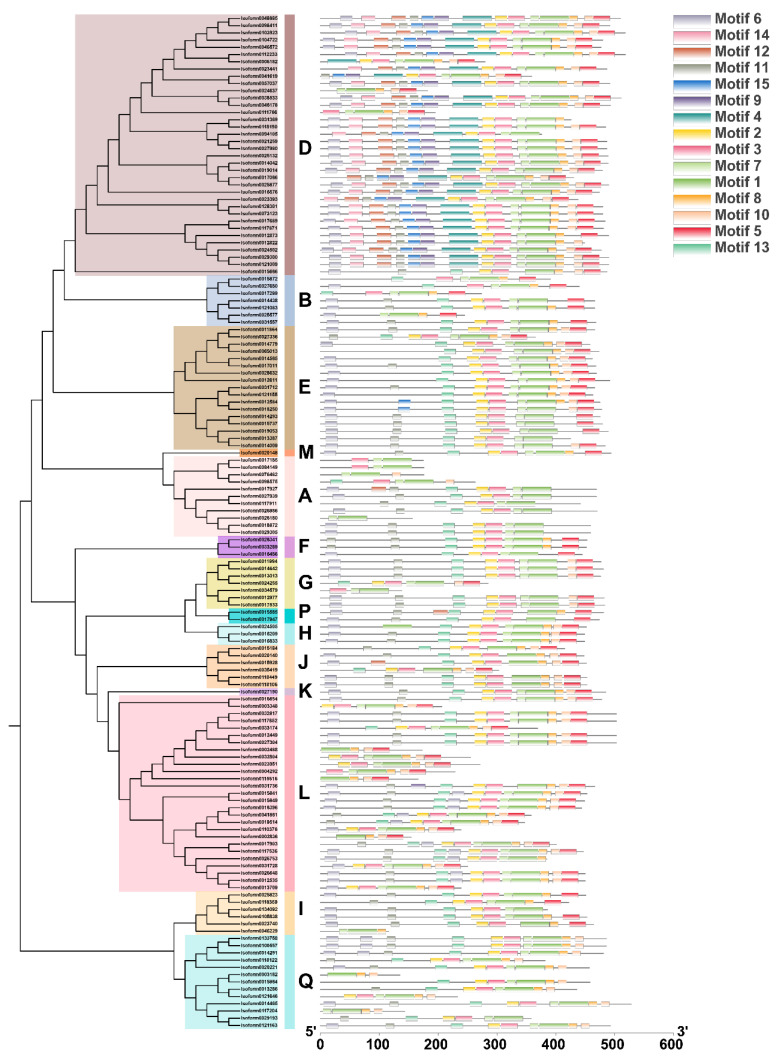
Distribution of conserved motifs of CaUGTs based on an evolutionary relationship. Phylogenetic tree and group classification of 140 CaUGTs on the left; 15 conserved motifs of CaUGT family members on the right. The sequence information for each motif is provided in Appendix A.

**Figure 4 molecules-26-04650-f004:**
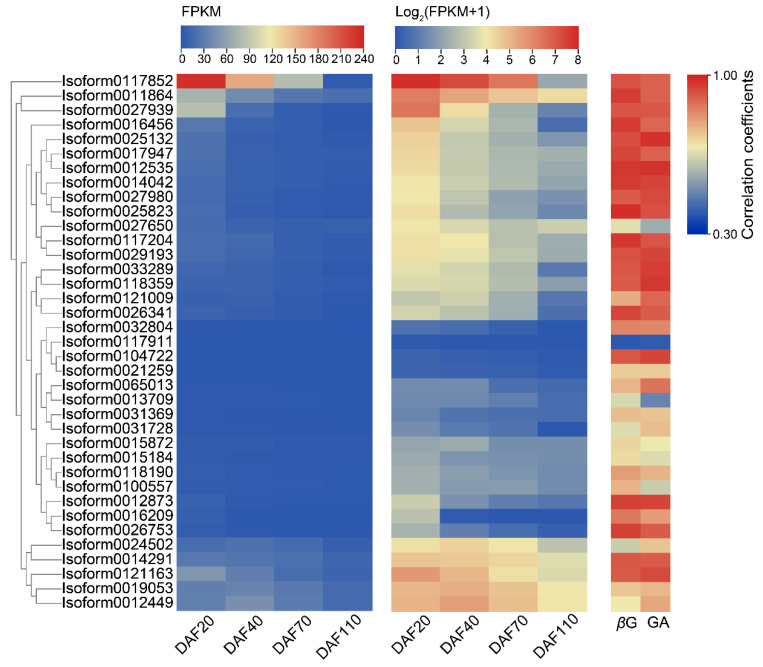
The expression patterns of *CaUGTs* in four developmental stages of *C. album* fruits and the correlation coefficients between *CaUGT* expression levels and GA and *β*G contents. FPKM values of four developmental stages days (20 days after flowering (DAF), 40 DAF, 70 DAF, and 110 DAF) of *C. album* fruit were obtained from RNA-seq data of our lab.

**Figure 5 molecules-26-04650-f005:**
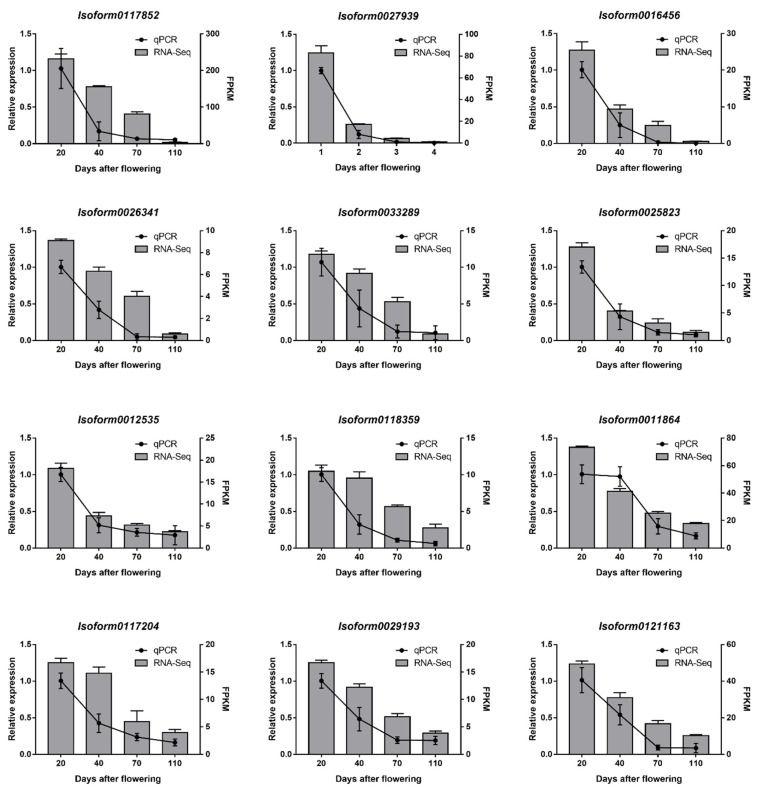
The relative expression of the *CaUGTs* in the developmental stages of *C. album* fruits. Relative expression of 12 genes was examined by the RT-qPCR and normalized with the reference gene *ACT7*. Each reaction was performed in three biological replicates with three technical replicates. Data are means ± SD. Error bars represent standard deviations.

**Figure 6 molecules-26-04650-f006:**
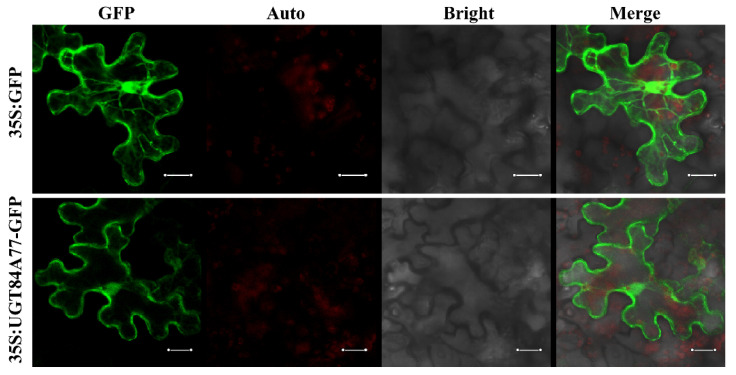
Subcellular localization of UGT84A77 in *Nicotiana benthamiana* leaves. GFP: GFP channel; Auto: chloroplast auto-fluorescence; Bright: light microscopy image; Merge: merged images of three channels. Scale bars represent 20 µm.

**Figure 7 molecules-26-04650-f007:**
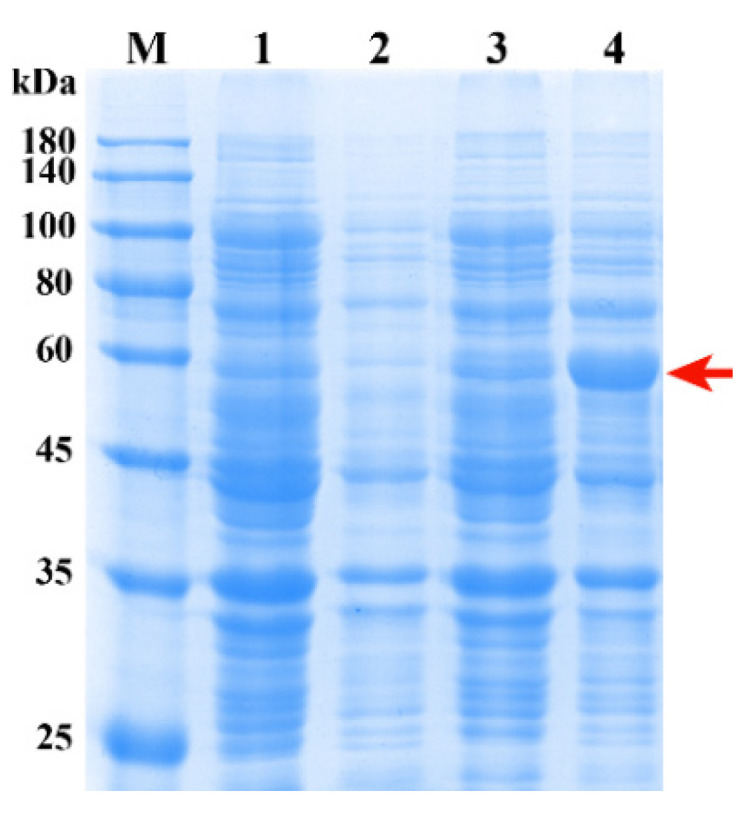
SDS-PAGE electrophoresis of prokaryotic expression of pET28a-*UGT84A77*. M, Protein Marker; lane 1, pET28a without isopropyl-*β*-d-thiogalactopyranoside (IPTG); lane 2, pET28a, induced by 0.2 mM IPTG; lane 3, pET28a-UGT84A77 without IPTG; lane 4, pET28a-UGT84A77, induced by 0.2 mM IPTG; the arrow indicates the expressed fusion protein, approximately 57 kDa.

**Figure 8 molecules-26-04650-f008:**
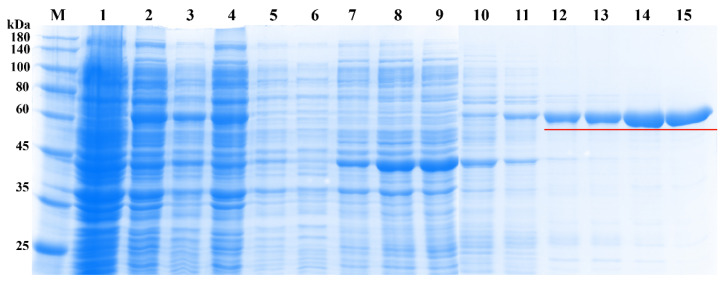
SDS-PAGE determination of purified prokaryotic expression protein pET28a-*UGT84A77*. M, protein marker; lane 1 indicates protein expression not induced by IPTG; lane 2 indicates total protein induced by IPTG; lane 3 indicates total soluble protein in supernatant; lane 4 indicates insoluble protein in pellet; lane 5 indicates effluent after flowing through resin; lane 6 indicates washing buffer after flowing through resin; lanes 7–15 indicate the proteins eluted by 40, 60, 80, 100, 120, 140, 160, 180, and 200 mM imidazole eluents, respectively; the underline indicates the purified objective protein of pET28a-*UGT84A77*.

**Figure 9 molecules-26-04650-f009:**
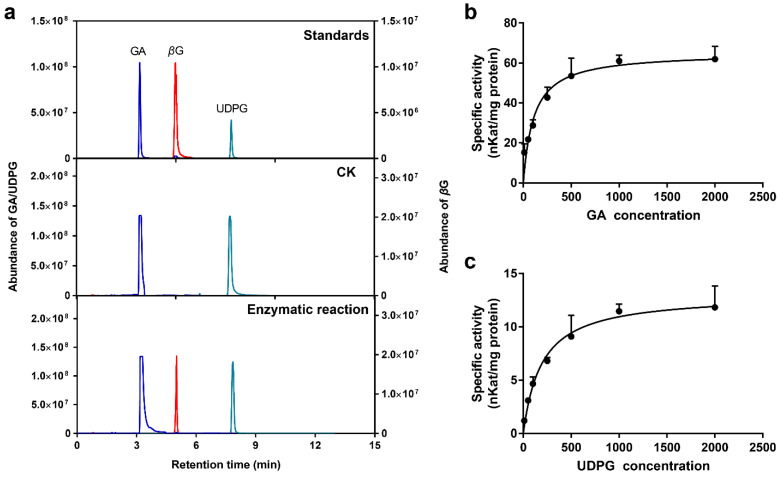
Enzymatic activity assays of recombinant UGT84A77 protein. (**a**) Identification of enzymatic products of UGT84A77 recombinant protein by UPLC-MS/MS; (**b**) the Michaelis–Menten curve obtained at different concentrations of GA; (**c**) the Michaelis–Menten curve obtained at different concentrations of UDPG.

**Table 1 molecules-26-04650-t001:** Kinetic parameters of purified UGT84A77 recombinant protein for GA and 5′-diphosphate glucose (UDPG).

Substrate	Vmax (nKat/mg)	Km (μM)	Kcat (s^−1^)	Kcat/Km (s^−1^·M^−1^)
GA ^a^	65.21 ± 3.22	108.90 ± 21.06	3.71 ± 0.18	34,076.64
UDPG ^b^	13.06 ± 0.66	193.30 ± 34.33	0.74 ± 0.04	3844.87

Note: ^a^ GA was used as a sugar acceptor; ^b^ UDPG was used as a sugar donor. Values are estimates ± Std. Error.

## Data Availability

The 140 CaUGTs datasets generated during the current study are not publicly available due the *C. album* full-length Iso-Seq transcriptome database are not public but are available from the corresponding author on reasonable request. Other data generated or analyzed during this study are included in this published article and its supplementary information files.

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
