# Peer review of "Identification and Characterization of Glucosyltransferase That Forms 1-Galloyl-β-d-Glucogallin in Canarium album L., a Functional Fruit Rich in Hydrolysable Tannins"

_molecules, 2021, doi:10.3390/molecules26154650_

Round 1
Reviewer 1 Report
The paper has a solid and valid aim, for which the authors should be acknowledged. However, there are some typos throughout the text (e.g., incorrect verbs, awkward sentences) that the authors should revise to improve readership. In relation to the scientific content, I have the following comments.
- Introduction: Please describe better which type of UGTs are associated with HTs.
- M&M: If the aim was to characterize the variety of UGTs (and HTs) why was only one variety and one locality used?
- Fruit samples were collected during several DAFs but no explanation is given.
- “According to the RNA-seq data from our laboratory (unpublished), CaUGTs, whose expression level changes were similar to that of metabolites, were selected as the candidate genes to analyse the expression patterns.” – this is a very circular information. How were genes selected? Based on what? And which type of genes?
- Results: the text is often mixed with what seems a discussion. For instance: “During the growth and development of C. album fruit, the dynamic change of total HTs contents was a down trend (Figure 1a). However, the HT content in ripened C. album fruits is higher than that in most fruits and traditional Chinese herbs [28,32,33]. There are many more examples in Results, which in some cases are even repeated in the discussion. The authors need to re-write those sections.
- “Fifteen conserved motifs from 140 CaUGTs were identified by MEME online software; almost all CaUGTs contained motifs 1, 2, 3, 7, and 8, and all the CaUGTs contained motif 1, which included the complete PSPG box (positions 4~47) (Figure 3, Figure S2). This indicated that the identification of CaUGT family members was reliable”. Why? The authors need to explain this much better.
- “Based on RNA-seq data, we chose 37 CaUGTs down-regulated during the developmental of C. album fruits as candidate UGTs, to further analyse the expression patterns. As shown in Figure 4, most the candidate CaUGTs showed a high correlation with βG content, and their trends of the relative expression were similar to that of βG contents.”. Again, if the authors already choose genes that were related, and of course, the correlation and expression should be high. As stated above this information is very hard to follow.
- “The results showed that the expression pattern of RNA-seq was reliable.” Why shouldn’t it be? Do the authors have any reason for suspecting this?
- The discussion is overall very well written but, as mentioned previously, some ideas were already expressed in Results. Basically, the main message is repetitive.
A major constrain that I see in this study is the fact that some results are based on unpublished data, for which the reader has no previous information, and therefore makes those experiments simply not reproducible. I understand that the authors are saying that they can provide the data upon request – but to be fair, we all know that sometimes you write to authors, and they simply do not respond. I suppose the authors are still working on that data but they can deposit it on NCBI and ask for a delay until everything is published. Please think that there is absolutely no information on this paper on how the authors reached the information related to the genes tested. What was the sequence platform used for Rna-seq? How were raw data analysed? How were data trimmed? Was a de novo or a reference assembly made? Are the results coming from genes or DEGs? This is obviously outside the scope of this paper, but it is the foundation of some data presented here, which might change substantially according to the questions that I have made. Please think about that.
Reviewer 2 Report
Ye et al report the identification of 140 UDP-glucosyltransferases (UGTs) in Canarium album L., their phylogenetic analysis and the partial characterization of one of them (UGT84A77) involved in hydrolysable tannins’ (HTs) biosynthesis. These are interesting molecules with many health/therapeutic potential applications. The paper is well organised however I have some suggestions.
Introduction section might be improved mentioning the mechanism(s) at the bases of beneficial effects of HTs.
Fig. 1A: Authors might indicate the ripening stage in the graph, or insert pictures of the fruit at each day indicated in the graph, I guess it would be helpful
L102: “176” is written twice, is it a typo?
L112-117: it is not clear. A total of 16 UGTs were included in the phylogenetic analysis, particularly 4 of P. persica, 3 of Z. mays and 1 of P. granatum were choosed to represent the O, P and Q groups, is it correct? Please modify concordantly figure 2 description (L128-133)
L120: “was consistent with of the results in” is it a typo?
L 131: Please, be consistent with terms. Use the latin name Z. mays
L 155: at least three stages, in the fourth stage it seems 0, is it correct?
Fig. 5: the 20 DAF qPCR data do not show standard deviation bars
L 188: the GenBank accession number has not been published yet
L 201: “liked the signal of GFP protein” is “liked” a typo?
L 207: Authors always refer to prokaryotic expression but never mention the host organism, I think that E. coli should be mentioned
L285: please change “product” in plural form
L291-294: several species are inserted for the first time, so, please, indicate the genera in the extensive form e.g., Pruns mume
L 299: May Authors indicate which is the difference among the 18 UGT groups?
L 342-348: Please insert references
L 356: “but more extensive substrates” please, modify in “but the study of more extensive substrates”
L 495: Did Authors used GA and UDPG in the reaction mix at the same time?
L 496: if Authors refer only to UGT84A77 "...proteins" become "...protein"
L 496-497: From what reported by authors, the enzymatic activity has been tested at pH 7.0 and 30°C. Considering the data in Fig. S4 I would have rather tried pH 5.0 and 20°C, why did authors choose such reaction conditions?
L 498: Please, insert “negative” control
L 499: “effects of pH 4.0 to 7.0 and temperature of 0°C to 40°C” "ranging from…to" or please, rephrase
L 504: Has the reaction mixture been lyophilized?
L 700: “Prunus Mume” please use the lower case for mume
Round 2
Reviewer 1 Report
I appreciate the authors efforts to overcome my doubts, which were erase in this present version.